



# Modeling the formation of toma hills based on fluid dynamics with a modified Voellmy rheology

Stefan Hergarten[1]

[1]Institut für Geo- und Umweltnaturwissenschaften, Albert-Ludwigs-Universität Freiburg, Albertstr. 23B, 79104 Freiburg, Germany

**Correspondence:** Stefan Hergarten
(stefan.hergarten@geologie.uni-freiburg.de)

**Abstract.** Toma hills are the perhaps most enigmatic morphological feature found in rock avalanche deposits. While it was already proposed that toma hills might emerge from the fluid-like behavior of rock avalanches, there still seems to be no consistent explanation for their occurrence. This paper presents numerical results based on a modified version of Voellmy's rheology, which was recently developed for explaining the long runout of rock avalanches. In contrast to the widely used original version, the modified Voellmy rheology defines distinct regimes of Coulomb friction at low velocities and velocity-dependent friction at high velocities. When movement slows down, falling back to Coulomb friction may cause a sudden increase in friction. Material accumulates in the region upstream of a point where this happens. In turn, high velocities may persist for some time in the downstream and lateral range, resulting in a thin deposit layer finally. In combination, both processes generate more or less isolated hills with shapes and sizes similar to toma hills found in real rock avalanche deposits. So the modified Voellmy rheology suggests a simple mechanism for the formation of toma hills.

## 1 Introduction

Toma hills are cone- to pyramid- or roof-shaped hills consisting mainly of landslide material (Abele, 1974). While toma hills seem to be more or less isolated, they are typically clustered. In the European Alps, such structures have been documented in the Rhine Valley close to Domat/Ems (Abele, 1974), at Fernpass (Prager et al., 2006), Almtal (van Husen et al., 2007), Obernberg (Ostermann et al., 2012), Eibsee (Ostermann and Prager, 2016), and Pragser Wildsee (Ostermann et al., 2020).

The formation of toma hills is still enigmatic. Results from dating (e.g., Ivy-Ochs et al., 2009; Ostermann et al., 2012) refute any effects of glaciation. Since the deposits of rock avalanches are typically hummocky, it makes sense to assume that toma hills are already formed during the emplacement and are not the result of a later modification. Ostermann et al. (2012) already discussed the occurrence of transverse ridges and toma hills in the Obernberg Valley in the context of the fluid-like behavior of rock avalanches. Beside the potential occurrence of waves and surges, the collapse of the fluid-like behavior below a minimum required kinetic energy played a part in this discussion.

However, there is still no direct evidence that toma hills emerge from the fluid-like behavior of rock avalanches. Several laboratory experiments on granular flow have been conducted with focus on rock avalanches (e.g., Pouliquen et al., 1997; Shea and van Wyk de Vries, 2008; Paguican et al., 2014; Valderrama et al., 2018), which were partly able to predict the occurrence





of hummocky topographies and ridges. Cone- or pyramid-shaped hills were, however, not found. The same holds for particle-based numerical simulations (e.g., Campbell et al., 1995; Mead and Cleary, 2015; Johnson et al., 2016) and for continuum simulations based on the fundamental theory proposed by Savage and Hutter (1989).

The lack of direct evidence from granular dynamics motivated alternative approaches. More and Wolkersdorfer (2019) revisited the idea of a long-term modification of the topography by internal erosion. Knapp et al. (2022) proposed a concept

involving an already existing lake, which may have been formed by a previous rock avalanche. This idea is, however, still on a qualitative level and appears to be site-specific.

This study takes up the idea that toma hills emerge from the complex, fluid-like behavior of rock avalanches. It starts from the modification of Voellmy's rheology (Voellmy, 1955; Salm, 1993) proposed by Hergarten (2024c). Voellmy's rheology was originally developed for snow avalanches, but is nowadays also applied widely to rock avalanches. In its original form,

however, it is not able to reproduce the long runout of large rock avalanches without assuming an artificially low coefficient of friction. The modified rheology reproduces the long runout reasonably well without assuming specific processes that reduce friction, such as frictional heating (Erismann, 1979; de Blasio and Elverhoi, 2008; Lucas et al., 2014) or acoustic fluidization (Johnson et al., 2016).

The existence of two distinct regimes of granular flow is the main idea behind the modified Voellmy rheology. The ideas

of Voellmy (1955) are adopted for high velocities, resulting in an effective friction that is proportional to the square of the velocity. In turn, Coulomb friction is assumed at low velocities. The velocity at which the transition occurs was derived from reinterpreting the concept of random kinetic energy (Buser and Bartelt, 2009; Bartelt and Buser, 2010). As a key point, friction may be much lower in the fast regime than in the slow regime. In this case, friction may increase suddenly when movement slows down and falls back to Coulomb friction. In this sense, the modified Voellmy rheology is a perfect starting point for

elaborating the ideas about a fluid-dynamical origin of toma hills discussed by Ostermann et al. (2012) further.

## 2 Approach

There was no specific strategy for selecting a suitable study site except for keeping it simple. As a first step, the model for rockslide disposition proposed by Hergarten (2012) was applied to a given topography, here the digital terrain model of Tyrol with a mesh width of 5 m (Land Tirol, 2023). This model yields a large number of potential detached volumes without taking

into account the existence and orientation of faults or any other properties of the rocks. The parameter values ($s_{\min} = 1$, $s_{\max} = 5$) were the same as used in the study of Argentin et al. (2021) focusing on landslide dams. The most recent version of the model was used, which also takes into account the local orientation of the failure surface. This modification avoids failure of entire mountains into multiple directions and thus reduces the number of obviously unrealistic detached volumes.

The Obernberg Valley was selected as a study site by scanning the spatial distribution of $10^6$ potential events visually. This

valley was already subject to a rock avalanche in the past, which was investigated in detail by Ostermann et al. (2012). Since the valley already contains toma hills, its morphology should in principle be suitable for generating toma hills. Figure 1 shows the topography including the predicted area of detachment (red), where the largest predicted volume was chosen. This volume is





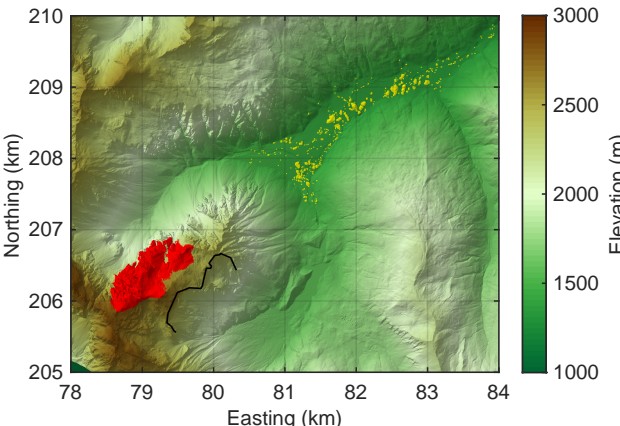

**Figure 1.** The study site in the Obernberg Valley. The red region depicts the assumed detachment area. The black line illustrates the scarp of the real rock avalanche (redrawn from Ostermann et al., 2012). Yellow patches are removed local maxima. Coordinates refer to EPSG:31254 (MGI Austria GK West).

about 80 million cubic meters, which is almost twice as large as the volume of the real rock avalanche estimated by Ostermann et al. (2012).

Since the assumed detachment area is in a different branch of the valley than the original event, the existing deposits can interfere with the simulated rock avalanche only in the lower part of the valley. While this part contains only a small fraction of the deposited volume, it hosts the toma hills. Therefore, existing hills were removed before simulating the rock avalanche. For this purpose, closed contour lines in the valley were computed in 0.25 m intervals. All points within these contour lines with an elevation higher than the contour line were replaced by minimal surfaces (yellow areas in Fig. 1). In combination with

the closed contour lines, this procedure is almost the same as clipping the areas inside the contour lines to the elevation of the respective contour line, but yields a slightly smoother transition at the edges.

  Version 2 of the model MinVoellmy (Hergarten, 2024b) was used for simulating the rock avalanche. As a modification of version 1 described by Hergarten (2024d), it uses thickness-weighted central difference quotients for the gradient of the surface. One-sided difference quotients towards the steepest descent are only used at local maxima. This version was already included

in the model intercomparison by Wirbel et al. (2024).

  The model MinVoellmy implements the modification of Voellmy's rheology proposed by Hergarten (2024c). Friction is represented by a shear stress of

$$
\tau = \begin{cases} \mu\sigma & \text{for} & v < v_{\mathrm{c}} \\ \frac{\rho g}{\xi} v^2 & & v \geq v_{\mathrm{c}} \end{cases} \tag{1}
$$

at the bed. The upper row describes Coulomb friction with a coefficient $\mu$, where $\sigma$ is the normal stress. The lower row describes

velocity-dependent friction as proposed by Voellmy (1955) originally for snow, where $\rho$, $g$, and $v$ are density, gravity, and vertically averaged velocity, respectively. The parameter $\xi$ refers to the roughness of the bed.



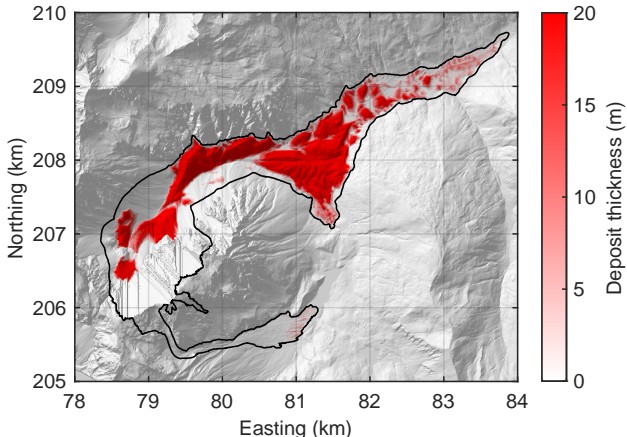

**Figure 2.** Final deposit thickness. The black outline encloses the region affected by the rock avalanche.

The crossover velocity $v_c$ depends on the thickness $h$ of the mobile layer. Hergarten (2024c) derived the relation

$$v_c \propto \sqrt[3]{\xi h} \tag{2}$$

by reinterpreting the random kinetic energy model (Buser and Bartelt, 2009; Bartelt and Buser, 2010), which describes the
supply of kinetic energy of random particle motion and its consumption. Equation (2) requires a factor of proportionality,
which is defined by $v_c$ at $h = 1$ m (at given $\xi$) in MinVoellmy. All occurrences of $\rho$ finally cancel since driving gravitational
forces and normal stress $\sigma$ are proportional to $\rho$. So MinVoellmy requires three physical parameters $\mu$, $\xi$, and $v_c$ at $h = 1$ m.

Several simulations were performed with different values of $\xi$ and $v_c$, while $\mu = 0.75$ was kept constant as a typical value
for Coulomb friction of rocks. Constant time increments of $\delta t = 0.01$ s were used, ensuring that the transport distance in each
step is much smaller than grid spacing and thus maintaining numerical stability according to the Courant–Friedrichs–Lewy
criterion. As a further technical setting, it was assumed that layers thinner than $1$ mm cannot move.

## 3 Results and discussion

### 3.1 Morphology of the deposits

The parameter combination $\xi = 500$ m s$^{-2}$ and $v_c = 4$ m s$^{-1}$ was selected for an in-depth analysis. Figure 2 shows the final
deposits, which extend further downstream by about $1$ km than the deposits of the real rock avalanche described by Ostermann
et al. (2012). Figure 3 shows a close-up of the deposits in the valley with contour lines of peaks with more than $3$ m prominence.
The respective outermost closed contour line defines the base for the height and inner contour lines are plotted in intervals of
$2$ m.





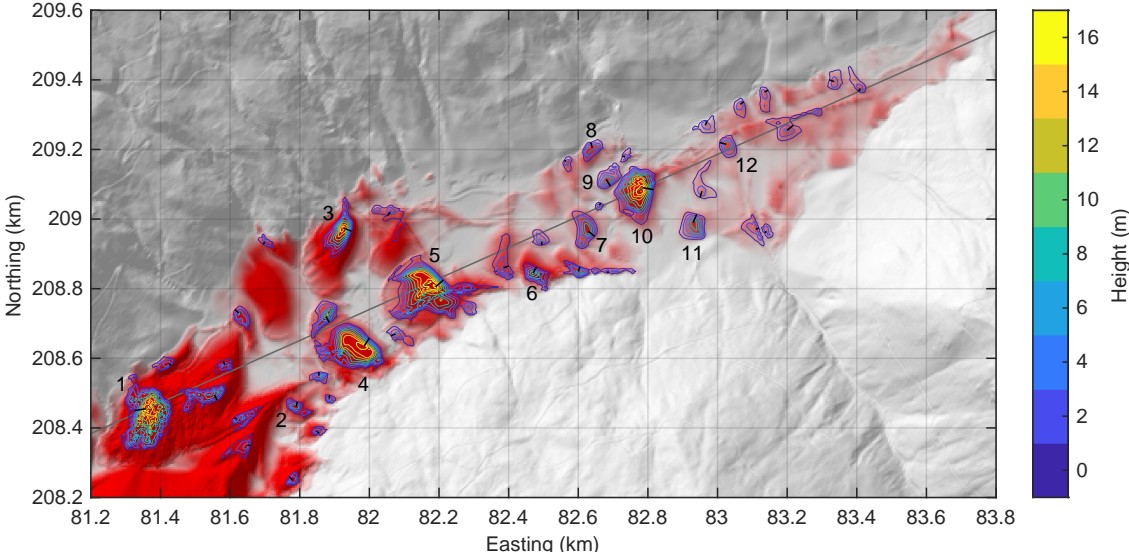

**Figure 3.** Morphology of the deposits in the valley. Colored lines are closed elevation contour lines drawn in 2 m intervals around hills with at least 3 m prominence, where the outermost contour line defines the base level. The red color describes the thickness of the deposits adopted from Fig. 2. Black and gray lines and labels refer to Figs. 5–7.

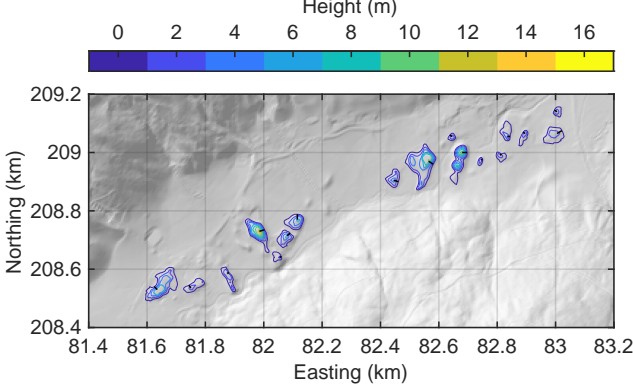

**Figure 4.** Hill structures in the real topography. Colored lines are closed contour lines drawn in 2 m intervals, where the outermost contour line defines the base level. Black lines refer to Fig. 5.

Figure 4 shows the results of the same analysis for the real topography. The biggest hills are obviously smaller than those obtained in the simulation. While the maximum prominence is about 19 m in the simulated deposits, it is only about 12 m in the real topography. Comparing the shapes in plan view is, however, difficult beyond the visual impression of some similarity.

For investigating the cross-sectional geometry, the prominence of each hill is considered in relation to two different radii. One of them is the minimum radius, defined by the shortest distance of the peak to the outermost contour line. It is illustrated





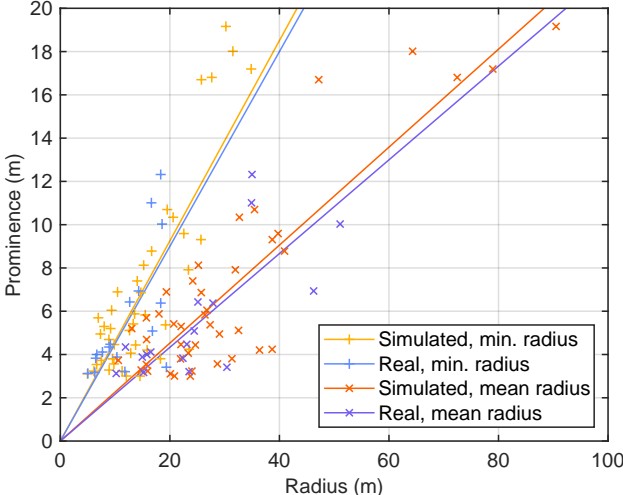

**Figure 5.** Prominence of hills in relation to minimum radius (black lines in Figs. 3 and 4) and mean radius. Straight lines are fitted linear functions through the origin.

by black lines in Figs. 3 and 4. The second radius is the mean radius, which is obtained by converting the outermost contour
line into a circle of equal area. Figure 5 shows the prominence of all hills vs. the two radii.

The straight lines were obtained by fitting linear functions through the origin. While individual hills scatter strongly around these lines, the lines themselves agree strikingly well between simulated and real hills. This means that the simulated hills have practically the same steepness as the real hills on average. The respective mean slopes are 0.46 into the steepest direction (minimum radius) and 0.23 on average (mean radius).

Figure 6 shows profiles of the numbered hills from Fig. 3 into the steepest direction. As a first observation, some profiles (3, 4, 6) contain a more or less straight segment that is considerably longer than the minimum radius illustrated by the black lines. This property is owing to the irregular shape of the hills and emphasizes the role of the prominence as a lower bound for the height of the respective hill. For the two steepest hills (3, 5), the slope of the straight segment is close to the coefficient of Coulomb friction, $\mu = 0.75$. This value defines the maximum stable slope. So some flanks of the hills are at the limit of
stability. The majority is, however, less steep, although also partly straight.

## 3.2 Are the hills isolated?

As a second characteristic property beyond the surface morphology, Abele (1974) described toma hills as isolated objects. However, the question what this means exactly is not trivial. While toma hills typically look like islands standing out of an alluvial plain, some clustering is often found. Ostermann et al. (2012) even found connected hills forming some kind of ridges
in the Obernberg Valley, which can partly be recognized in Fig. 4. These observations raise the question whether toma hills are typically isolated patches of rockslide material or outcrops of a continuous layer in the subsurface. Resistivity measurements in





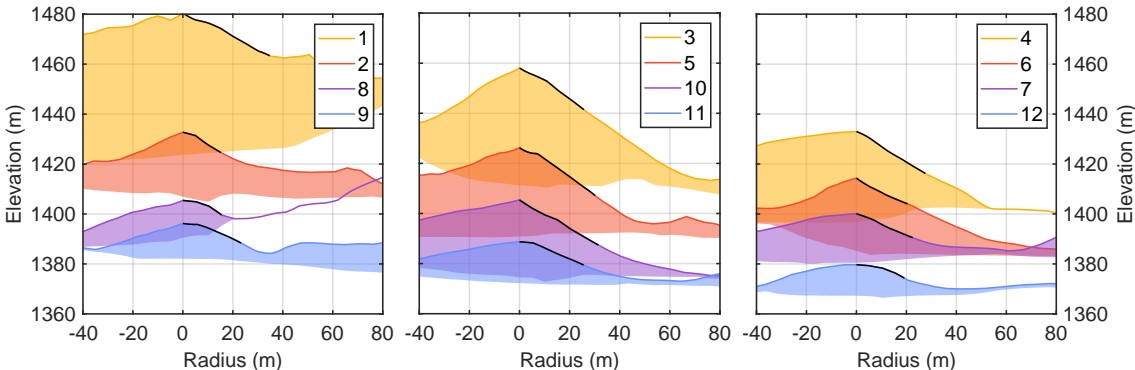

**Figure 6.** Profiles across the 12 numbered hills from Fig. 3 into the direction of the shortest radius. Black lines refer to the black lines shown in Fig. 3. Negative values of the radius point into the opposite direction. Filled areas illustrate the thickness of the deposits.

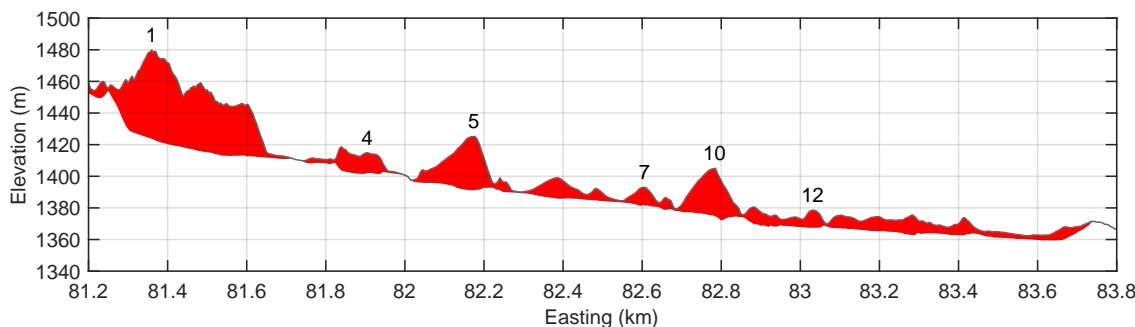

**Figure 7.** Longitudinal valley profile drawn through hills 1 and 10. For a better identification in Fig. 3, easting is plotted instead of length along the profile. Elevation is exaggerated four times.

the Rhine Valley by Knapp et al. (2022) could not answer this questions uniquely, but suggest that there may be some rockslide material beneath the alluvial deposits.

As already visible in Fig. 3, the hills generated by the model are not just hummocks at the surface of a thick layer of rockslide material. There are even areas between the hills with very little thickness. In turn, however, several hills are not clearly separated. Some of the connections are like narrow bridges. Revisiting Fig. 6 confirms that the thickness between the hills is irregular. Hills 4 and 8 approach a very small deposit thickness into their steepest direction. For the majority of the hills, however, a thickness of some meters persists at the foot.

Figure 7 shows a longitudinal valley profile drawn through hills 1 and 10. Hill 4, which is crossed at a saddle between two local maxima, looks quite isolated in this direction. Hills 5 and 10 are not isolated in the profile direction, but are much higher than the minimum deposit thickness between the respective hill and its neighbors. In turn, the lower part of the profile is characterized by a more continuous deposit layer, which is partly related to the road.





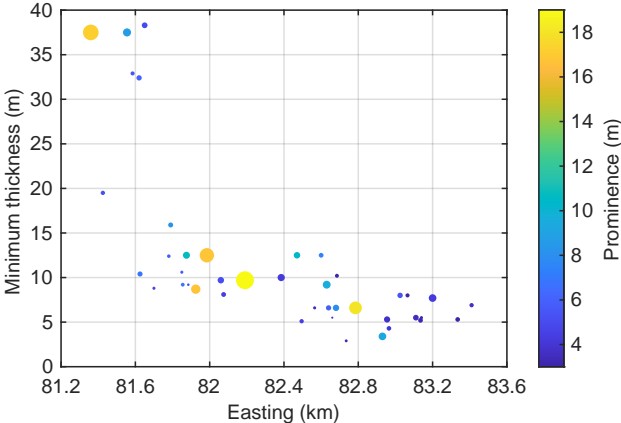

**Figure 8.** Minimum deposit thickness around each hill. Colors refer to prominence and marker sizes to the area of prominence, defined by the area within the respective outermost contour line in Fig. 3.

Formally, however, the hills are not separated clearly in terms of deposit thickness. Figure 8 shows the minimum thickness around each hill, defined by the lowest thickness contour line around each hill that does not include another hill. This means that the respective hill would be isolated completely if we removed this thickness. The lowest thickness is 2.9 m, and only 3 out of the 42 hills are separated by a thickness lower than 5 m. This separation is weaker than it could be expected from the previous considerations. It is, however, related to the complex pattern of the deposits already recognized in Fig. 3. Some hills are quite close to each other and are thus not separated clearly, but narrow, ridge-like connections also contribute to the weak separation.

Overall, the model predicts more or less isolated hills. This finding is not inconsistent with previous knowledge about toma hills. Taking into account that the uppermost rockslide deposits may be reworked by fluvial processes or even anthropogenically, validating the results of the model quantitatively may, however, be challenging.

### 3.3 The mechanism of formation

The formation of hills is closely related to the discontinuity in friction at the velocity $v_c$ in the model. If the fluid-like behavior collapses at any point, friction increases suddenly, which causes a rapid decrease in velocity and finally some kind of traffic jam upstream of the respective point.

Figure 9 illustrates at which time the fluid-like behavior collapses in a part of the valley. Overall, this collapse proceeds downstream, owing to the low friction for $v \geq v_c$. If friction was high, particles would be decelerated strongly when moving downstream, which could let the collapse start at the front and propagate upstream. Then the entire mass could accumulate in a small area.

Since this is not the case here, the formation of hills hinges on local variations in the time of collapse. As expected, hills are formed in Fig. 9 preferably in zones of an early collapse or slightly upstream. Local minima in the time of collapse are



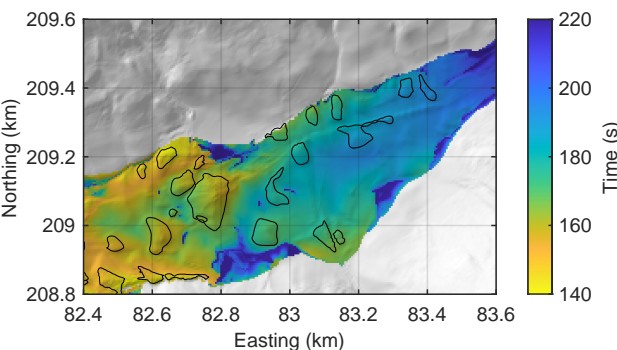

**Figure 9.** Time of collapse of the fluid-like behavior. Black lines are the outlines of the hills shown in Fig. 3.

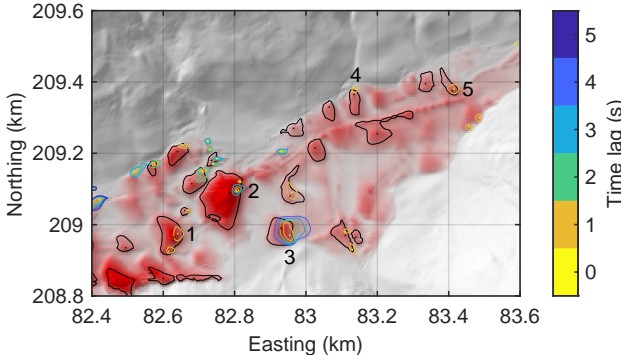

**Figure 10.** Local minima in the time of collapse of the fluid-like behavior with a (negative) prominence of at least 1 s. Contour lines are drawn around the minima in intervals of 1 s. Black lines are the outlines of the hills shown in Fig. 3 and black dots show the respective nearest peaks in topography. Labels refer to Fig. 11.

particularly relevant here. The yellow points in Fig. 10 are such local minima with a (negative) prominence of at least 1 s. So the collapse spreads from these points into all directions for at least 1 s.

The solid lines in Fig. 11 show the velocity and the thickness of the 5 seed points labeled in Fig. 10. The collapse of the fluid-like behavior takes place in a phase of decreasing velocity. However, this does not imply that the particles are already decelerated much since the plot does not follow moving particles, but refers to the particles at a given position (Eulerian description). Friction is still low as long as $v \geq v_c$, and the decrease in velocity arises from the ceasing supply of material from the upstream region.

After falling back to Coulomb friction, however, the particles are decelerated strongly. Assuming a horizontal bed and a horizontal surface, this deceleration is $\mu g \approx 7.3$ m s$^{-2}$, which lets the particles completely stop in less than $1.4$ s from $v = 10$ m s$^{-1}$ (the typical order of magnitude in Fig. 11a) with a traveling distance of less than 7 m. In turn, the change in thickness at the seed points is not unique. It may increase or decrease, but there is typically little change.





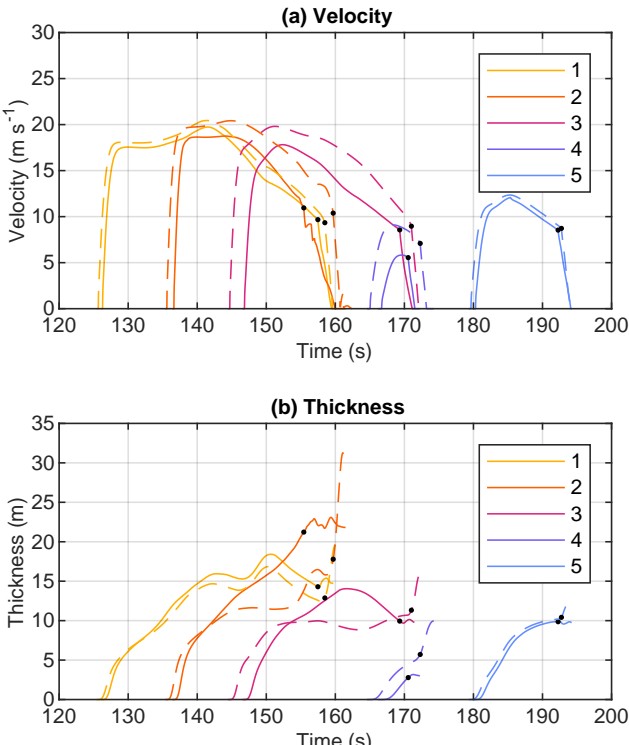

**Figure 11.** Velocity and thickness at the 5 hills marked in Fig. 10. Solid lines refer to the seed points (yellow points in Fig. 10) and dashed lines to the respective local maxima in elevation (black points in Fig. 10). Dots depict the collapse of the fluid-like behavior.

The dashed lines in Fig. 11 show the points that will become the respective local maxima in elevation (black points in

Fig. 10). By definition, these points fall back to Coulomb friction later than the respective seed points. Owing to their location upstream of the seed points, material accumulates here, causing an increase in thickness until the material comes to rest. While the thickness increases almost by a factor of two at peaks 2 and 4, there is almost no increase at hill 5. The majority of the peaks not shown here is also in the range between almost no increase and a factor of two. So the accumulation of material alone cannot explain the occurrence of more or less isolated hills. There must also be a decrease in thickness in the surrounding area.

Figure 12 illustrates the velocity field around the seed point 1. Time is measured relative to the time at which the seed point falls back to Coulomb friction ($t_0 = 157.48$ s). Flow continues with little changes in velocity for some seconds in the region upstream of the seed point, resulting in the accumulation described above. Some slight changes in the direction of velocity indicate that particles try to bypass the obstacle, but this effect is weak. Most of the incoming flux still moves towards the obstacle and lets it grow rapidly. Downstream, the opposite effect is observed. Particles that fell back to Coulomb friction are

rapidly decelerated, while particles that are already ahead keep moving at a high velocity. This difference leads to a depletion of material downstream of the obstacle and thus to a region with a low deposit thickness.




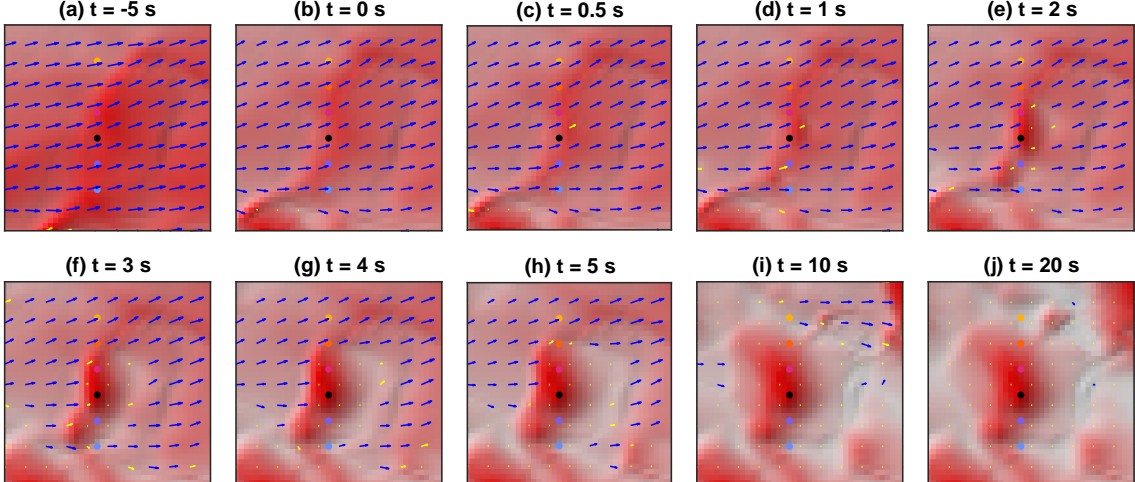

**Figure 12.** Velocity field around the seed point 1 from Figs. 10 and 11. Time is measured relative to the time when the seed point falls back to Coulomb friction ($t_0 = 157.48$ s). The domain is centered to the seed point and $200 \times 200$ m large. Blue arrows refer to velocity-dependent friction ($v \geq v_\mathrm{c}$) and yellow arrows to Coulomb friction ($v < v_\mathrm{c}$). Dots show the points considered in Fig. 13.

The behavior concerning the lateral direction is less systematic. Figure 13 illustrates the change in thickness for 6 points. The point that will become the peak (black line) is the first to fall back to Coulomb friction and experiences the strongest increase in thickness. The points shifted by $\delta y = \pm 25$ m are not much later, but behave differently. One of them ($\delta y = 25$ m) behaves similarly to the peak, but the southward point ($\delta y = -25$ m) even decreases in thickness. It is recognized in Fig. 12 that material accumulates rather upstream of this point. The point at $\delta y = 75$ m stays at $v \geq v_\mathrm{c}$ even almost 10 s longer. During this time span, thickness decreases from more than 8 m to less than 2 m and then further to about 1 m until movement stops completely. This depletion of material is not necessarily related to the dynamics at the hill. Here it is the effect of ceasing supply from the upstream region, making the layer thinner until it finally comes to rest. In turn, laterally elongated hills or even transverse ridges may occur if the collapse of the flow propagates laterally before the supply ceases.

So the formation of more or less isolated hills involves two components. The top of the hill is typically formed by accumulation of material, which starts from a point that falls back to Coulomb friction and propagates upstream. As a second contribution, material is depleted in the domain around the hill. This depletion is only partly related to the formation of the hill. The hill immediately suppresses the supply of material to its downstream range, leading to a rapid depletion here. In turn, fast flow may continue for a longer time in the upstream and lateral range, leading to a depletion of material just by the cessation of the incoming flux. Since this contribution depends on what happens in the upstream range rather than on what happens at the hill, it seems to be responsible for the observed large variation in shape of the hills.



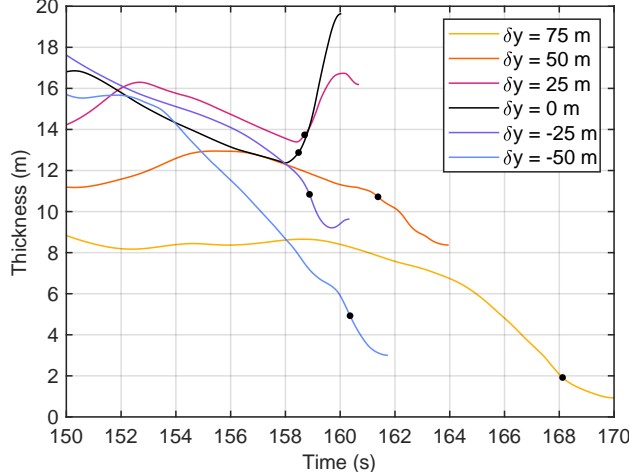

**Figure 13.** Thickness at the points marked in Fig. 12. The black line corresponds to the point that becomes the peak. Colored lines refer to points shifted into the north-south direction in steps of $25$ m. Black dots describe the time at which the fluid-like behavior collapses.

## 3.4 Influence of the parameters

The results obtained so far were obtained from a combination of parameter values that allows for a quite long runout in the valley. Figure 14 shows two scenarios that yield a shorter runout. Friction in the fluid-like regime was increased by a factor of 2 ($\xi = 250 \text{ m s}^{-2}$ instead of $500 \text{ m s}^{-2}$) in Fig. 14a. The other version (Fig. 14b) assumes the original friction, but an earlier collapse of the fluid-like behavior ($v_c = 5 \text{ m s}^{-1}$ instead of $4 \text{ m s}^{-1}$ for $h = 1$ m). The two parameter sets are related to each other by the minimum amount of random kinetic energy required to maintain the fluid-like behavior. According to the considerations of Hergarten (2024c), this amount is proportional to $\frac{v_c^3}{\xi}$. It is thus almost identical for the two parameter sets and twice as high as for the original scenario.

Runout is by more than 1 km shorter for both scenarios than in the original simulation (Fig. 3) and also shorter than that of the real rock avalanche in the valley. The two scenarios are not only similar concerning the runout, but also concerning the morphology of the deposits. Both versions produce hills, although over a shorter section of the valley, owing to the shorter runout. Some hills seem to be even at similar locations in both scenarios.

The profiles shown in Fig. 15 confirm the similarity of the morphologies, although the version with the increased friction (Fig. 14a) produces thicker deposits along large parts of the profile than the version with an earlier collapse of the fluid-like behavior (Fig. 14b). In some sections of the profile, the morphology even looks similar to that of the original scenario (Fig. 3).

These results suggest that the of minimum amount of random kinetic energy required to maintain the fluid-like behavior controls how far the rock avalanches travels along the valley. It therefore defines the potential to form toma hills. However, it seems that the formation of hills in the part of the valley covered by rock avalanche deposits is not very sensitive to the parameter values. The crucial condition seems to be that friction in the fluid-like regime must be much smaller than Coulomb friction.



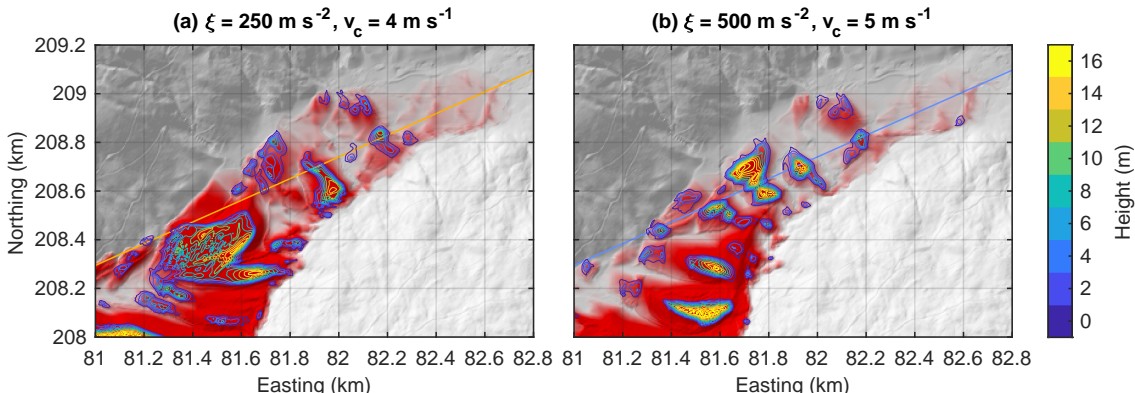

**Figure 14.** Deposits in a part of the valley obtained for two different parameter combinations. The value of $v_c$ refers to a thickness of $h = 1$ m. The scheme of colors and contour lines is the same as in Fig. 3. Straight lines correspond to the profile considered in Fig. 7.

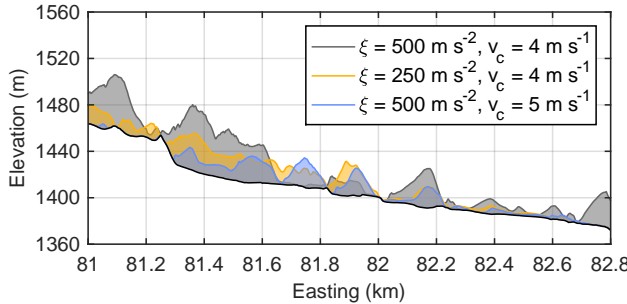

**Figure 15.** Profiles along the straight lines in Fig. 14 compared to the original simulation (Figs. 3 and 7). Elevation is exaggerated four times.

# 4   Conclusions and outlook

Based on the numerical simulation of a hypothetic rock avalanche, it was found that the modification of the widely used
Voellmy rheology proposed by Hergarten (2024c) is able to reproduce the occurrence of distinct hills in the deposits. The key point of the modified rheology is a sudden drop in friction at the transition from Coulomb friction to a fluid-like behavior. While this property was originally introduced in order to explain the long runout of rock avalanches without specific assumptions, it turned out to be responsible for the formation of hills, too.

The mechanism of forming hills involves two components, starting from falling back to Coulomb friction at one point. The
sudden increase in friction decelerates the particles strongly, causing material to accumulate in the domain upstream of the seed point. In turn, particles downstream of the seed point still move at high velocities, causing a rapid depletion of material. However, the formation of distinct hills requires a depletion also in the lateral region. This happens if flow at low friction continues there until supply ceases. In turn, hills become laterally elongated or even form transverse ridges if the fluid-like behavior collapses in the lateral region before supply ceases.



Since the shape of the simulated hills is quite irregular, a quantitative comparison with real toma hills is difficult. For the considered valley, however, the steepness of the simulated hills is strikingly similar to the steepness of existing toma hills generated by a real rock avalanche. This holds for the average steepness as well as for the slope into the steepest direction. For some of the simulated hills, the slope into the steepest direction is close to the maximum stable slope, which is defined by the coefficient of Coulomb friction. The majority of the hills is, however, less steep.

The simulated hills are more or less isolated in the sense that there are areas with little deposit thickness between them. In turn, there are also connections between hills with several meters of deposits. So the hills are formally not clearly separated in terms of deposit thickness. Although real toma hills often look like islands in an alluvial plain, the partial connection by rock avalanche deposit is not inconsistent with observations. Knowledge about the subsurface structure around toma hills is still incomplete. Validating or refuting the model requires more simulations and a more detailed analysis as well as more research 230 on real toma hills.

The occurrence of hills that look much like toma hills seems not to rely on specific parameter values. Concerning the parameters, it only seems to be important that friction in the fluid-like regime is much weaker than Coulomb friction. As there is even some similarity in the spatial pattern of the hills for different parameter values, the shape of the valley and the morphology of the valley floor also seem to be important. Finding out which valley morphologies facilitate the formation of 235 toma hills could be subject of a subsequent study.

So far the modified Voellmy rheology has been supported mainly by its ability to explain the long runout of rock avalanches and it dependence on volume (Hergarten, 2024c). The finding that it also predicts the formation of hills that look much like toma hills provides further support for this rheology.

*Code and data availability.* All codes are available in a Zenodo repository at https://doi.org/10.5281/zenodo.10932346 (Hergarten, 2024a). 240 This repository also contains the data obtained from the numerical simulations. Interested users are advised to download the most recent version of the MinVoellmy software from http://hergarten.at/minvoellmy (Hergarten, 2024b).

*Video supplement.* Two videos showing the simulation are available at http://hergarten.at/minvoellmy/tomahills (Hergarten, 2024b).

*Competing interests.* The author declares that there is no conflict of interest.



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
