# Peer review of "Modeling the formation of toma hills based on fluid dynamics with a modified Voellmy rheology"

_EGUsphere, 2024_

## Author Response (AR1)

Dear Reviewers, dear Editor,

thank you very much for your constructive comments! I am happy about the overall very positive reception. The points addressed in the reports are discussed below, where changes to the manuscript are highlighted in bold letters. Line numbers refer to the version with highlighted changes. In addition, I simplified Fig. 15 a bit in order to facility the comparison.

Best regards,

Stefan Hergarten

**Reviewer 1**

*This paper suggests that toma hills are formed when movement slows down increasing friction (Coulomb friction) causing an accumulation of material on the material source direction (upstream side) of a point at which this happens. High velocity may continue on the downstream and lateral sides and this results in thin deposits and more or less "isolated" toma hills.*

*I am reviewing as an expert on how hummocks are formed in debris avalanche deposits based on analogue granular experiments.*

*My full review is as follows:*

*Does the paper address relevant scientific questions within the scope of ESurf? The paper suggests a mechanism of forming toma hills (hummocks in volcanic debris avalanches), on the discussion of physical processes shaping the Earth's surface using numerical modelling; thus, within the scope of Esurf journal.*

*Does the paper present novel concepts, ideas, tools, or data? This work uses a modified Voellmy's rheology. This idea has two distinct regimes of granular flow; (1) the original Voellmy rheology adopted for high velocities that result in an effective (velocity-dependent) friction proportional to the square of the velocity and (2) Coulomb friction for low velocities where friction may be lower in fast regime than slow regime and velocity at transition was an interpretation of the concept of random kinetic energy. In this regard, this work presents a novel concept.*

*Are substantial conclusions reached? Yes, toma hills are reproduced from numerical simulations when there is a significant decrease in friction from original Voellmy rheology to Coulomb friction causing an accumulation of material while adjacent sides (downstream and lateral sides) continue moving at high velocities.*

*Are the scientific methods and assumptions valid and clearly outlined?*

*Yes, assumptions are outlined. Although, this paper assumes that Voellmy rheology and Coulomb Friction are known to the reader. It would be beneficial to the reader if a brief in-depth description of these would be presented in this work.*

**I added a short explanation of Voellmy's original rheology and how the modified version differs (lines 77–79),** but still prefer not to explain Coulomb friction explicitly.

*A clear distinction between, if any, between toma hills and hummocks would also widen the audience of this work with applicability not only for rock avalanches but also for volcanic and non-volcanic debris avalanches.*

To my understanding, toma hills are particular hummocks with quite low deposit thickness between the hummocks. In contrast, the majority of the observed hummocks seem to sit on rather thick deposits. I retreated to toma hills because all previous numerical and laboratory models I am aware of run into problems here. The modified Voellmy rheology also produces hummocks on thicker deposits, but I cannot be sure that it is better than existing approaches here. So I feel that an extension towards more general hummocks would result in losing the focus a bit.

*Line 48–49: It is not clear what this sentence means. A brief description of Hergarten (2012)'s rockslide disposition will be helpful. In this paragraph, it says the model does not consider faults and rock properties but in the same paragraph, it also says, the most recent version of the model was used to account for local orientation of failure surface. A confirmation of how much consideration was accounted for on the original source area geomorphology would make this clear.*

**I added a bit of explanation about the model and tried to clarify more explicitly that defining the source area is not subject of this study (lines 49–56 and 63–66).**

*In line 55: As the study area has been impacted by rock avalanches in the past, how will previous avalanches' depositional surface affect the current surface simulation (roughness of bed)? I assume that if a toma already exists in the current avalanche pathway, it will affect the emplacement and runout of current avalanche? How would volume affect the result of this work's simulations?*

In principle, this was already mentioned in lines 60–66. **I extended this part a bit (lines 67–73).** Beyond the removed existing toma hills it is, however, difficult to say anything about the roughness 8600 years ago compared to the present-day roughness including infrastructure etc.

*The simulation has been applied assuming a valley-filling rockslide debris avalanche. This work would be of interest to a wider audience, for example, researchers looking at volcanic and non-volcanic debris avalanches if a simulation could also be done on a topography without topographical barriers such as an adjacent elevated area. This would chase out how the impact on the the transition and changes in velocity as avalanche material is spreading freely.*

I already performed several simulations, but just inspected to deposit morphology visually before writing the manuscript. It indeed looks as if flowing along a valley with a quite flat floor promotes the formation of toma hills compared to scenarios with free radial spreading. However, it is not so trivial and I need to understand better why this is the case before writing anything about it.

*Are the results sufficient to support the interpretations and conclusions?*

*Several simulations were conducted, it is not clear why a bed roughness of 500 ms$^{-2}$ and factor of proportionality of 4 ms$^{-1}$ was chosen for an in-depth analysis and how changing these parameters affect the model.*

Since the modified Voellmy rheology is quite new and the value of $\xi$ cannot be adopted directly from simulations with the conventional rheology, there is still little knowledge on the values of these parameters. For this reason, a preliminary sensitivity analysis was presented in Sect. 3.4. **I added a remark (lines 102–103).**

*Is the longer runout in Figure 2 using the model due to the volume that is also ×2 that of Ostermann et al (2012)?*

Using the empirical relation introduced by Scheidegger (1973) ($H/L \propto V^{-0.16}$), the increase in volume should yield a 10 % longer runout, which could indeed explain the difference to some degree. However, real-world data scatter strongly around this relation, and the different flow paths may also have a big influence as well as the parameter values. So it seems not to be possible to give a clear reason for the longer runout.

*The biggest hills in the simulation is bigger than the real toma hills. How does the total number of toma hills and distribution compare between both simulation and reality?*

Originally, I wanted to stay on a more qualitative level and did not include any results about the size distribution. It seems that the prominence follows an exponential distribution (see figure) with little difference between simulation and real topography, but I am uncertain about its interpretation. **I added some results about the distributions in (lines 109–111 and 118–119),** but without going into details.

[Figure]

*Is the description of experiments and calculations sufficiently complete and precise to allow their reproduction by fellow scientists (traceability of results)? I appreciate the availability of matlab codes, to recreate figures and simulations. The results can be traced if you use the same data input as described in the text, which is good. It is however, imperative, for the codes to have an accompanying better documentation (as comments on the code or an accompanying read me file) to define each element in the code. An as example, fs, lw, cm in figure codes, and uhs, vhs, and others in simulation code are not defined. The article, if published, would immensely benefit if other researchers will understand the code and use their own DEM with the code.*

**I updated the repository,** although almost all of the variables that were not explained refer to properties of the diagrams (font size, line with, color map, . . . ). However, I am not sure how much it will bring since at least some of the analyses behind the figures are quite specific and I do not expect that anyone will repeat exactly the same procedure with different data.

*Do the authors give proper credit to related work and clearly indicate their own new/original contribution? Yes, proper credit was given on related work and new and original contributions of this work are clearly indicated.*

*Does the title clearly reflect the contents of the paper? Yes, the title clearly reflects the contents of the paper, which is on modelling the formation of toma hills based on a fluid dynamics using a modified Voellmy rheology.*

*Does the abstract provide a concise and complete summary? Yes.*

*Is the overall presentation well structured and clear? Yes.*

*Is the language fluent and precise? Yes.*

*Are mathematical formulae, symbols, abbreviations, and units correctly defined and used?*

*$s_{\min}$ and $s_{\max}$ are not defined in this work, although it points to Agentin et al. (2021), would be useful to say this here.*

**I added a very short explanation (lines 51–54),** but I feel that going into depth would be more distracting then enlightening here.

*Would be better to put $v_c$ on page 3, where the others are also described.*

Sorry, but defining where page or column breaks will be in the final paper is out of my influence.

*Should any parts of the paper (text, formulae, figures, tables) be clarified, reduced, combined, or eliminated? Figures 3 and 4 if they are put either side by side (or top and bottom) to make them easier to compare between simulation and real toma hills.*

I originally decided not to join these figures because Fig. 3 covers a larger range than Fig. 4 and is more important. For this reason, Fig. 3 should be large (two columns), while Fig. 4 can be small. A direct comparison is less important.

*Are the number and quality of references appropriate? Yes, but I would suggest looking at a few more as these might help support, or expand discussions in this work:*
*Kelfoun, K., & Druitt, T. H. (2005). Numerical modeling of the emplacement of Socompa rock avalanche, Chile. Journal of Geophysical Research: Solid Earth, 110(12), 113. https://doi.org/10.1029/2005JB003758*
*Thompson, N., Bennett, M. R., & Petford, N. (2010). Development of characteristic volcanic debris avalanche deposit structures: New insight from distinct element simulations. Journal of Volcanology and Geothermal Research, 192(34), 191200. https://doi.org/10.1016/j.jvolgeores.2010.02.021*

**I added the second paper to the reference to particle-based simulations (line 26).** The first paper might have been useful as a reference in my first paper about the modified Voellmy rheology, but was not aware of it at that time. However, I did not find a good way to bring it into the context of the recent manuscript.

*Is the amount and quality of supplementary material appropriate? Codes need to have better documentation to as explained in item 6 above to make it more useful for those who want to apply this using another DEM. In Figure6.m code, it is hard to see what to change to be able to reproduce the graph for toma hills, 3,5,10, 11 (center, Fig 6) and 4,6,7,12 (right, Fig 6).*

**I updated the repository.** However, there is nothing to be changed in figure6.m for the different subplots. The script just produces 3 files figure61.pdf, figure62.pdf, and figure63.pdf.

**Reviewer 2 (Martin Mergili)**

*The author presents an approach to numerically reproduce toma hills (more or less isolated hills in the distal area of rock avalanche deposits). For this purpose, he uses the relatively simple and straightforward Voellmy approach, which builds on bed friction and turbulent friction. Reinterpreting the idea of the random kinetic energy (e.g., Buser and Bartelt, 2009), he defines a flow thickness-dependent threshold velocity above which the bed friction does not act. Using this approach, he simulates a generic rock avalanche in the Obernberg Valley, Tyrol, Austria, and compares the resulting toma hills with those produced by a prehistoric event in the same area. The results are plausible, and a strong dependency of the formation of toma hills on the local topography is revealed.*

*This topic is of high scientific interest and significantly contributes to the ongoing scientific debate on the formation of toma hills. The discussion paper is clearly within the scope of the Earth Surface Dynamics journal. It is very well written, structured, and illustrated. Appropriate references are given to previous work, and the method and results are described and discussed in a clear and comprehensive way. I would definitely like to see this work published in Earth Surface Dynamics.*

*One aspect I thought about when reading the results and discussion section is the influence of the spatial resolution on the model results, and whether there would be some maximum cell size (in relation to the toma hill size) beyond which the formation of toma hills is blurred in the simulation. From my point of view, it is not mandatory to do some additional simulations with varying cell size within this publication. Such an exercise could also be a possible direction for follow-up research.*

The spatial resolution is indeed a problem in such simulations, and it is even worse than just blurring the hills at low resolution. The scenario of a long runout in an almost flat valley is extremely sensitive to the spatial resolution, and the threshold-like modified Voellmy rheology presumably even amplifies the problem compared to the original version. I think it is a general problem of depth-integrated models since small-scale oscillations of the fluid surface result in an acceleration of the entire column down to the bed. A detailed analysis including potential solutions would indeed be a good starting point for follow-up research. **For the moment, I included the results of a simulation with 10 m grid spacing instead of 5 m (lines 230–238 and Fig. 16)** in order to illustrate that the spatial resolution has a stronger effect on runout length than on the formation of the hills, similar to the discussion of the parameter values.

*Therefore, I recommend acceptance of the paper for publication in Earth Surface Dynamics.*